# CcMYB12 Positively Regulates Flavonoid Accumulation during Fruit Development in *Carya cathayensis* and Has a Role in Abiotic Stress Responses

**DOI:** 10.3390/ijms232415618

**Published:** 2022-12-09

**Authors:** Yige Wang, Hongyu Ye, Ketao Wang, Chunying Huang, Xiaolin Si, Jianhua Wang, Yifan Xu, Youjun Huang, Jianqin Huang, Yan Li

**Affiliations:** State Key Laboratory of Subtropical Silviculture, Zhejiang A&F University, Lin’an District, Hangzhou 311300, China

**Keywords:** *Carya cathayensis*, CcR2R3-MYBs, CcMYB12, flavonoid biosynthesis, gene expression, abiotic stress response, self-activation activity, transcriptional regulation

## Abstract

Flavonoid, an important secondary metabolite in plants, is involved in many biological processes. Its synthesis originates from the phenylpropane metabolic pathway, and it is catalyzed by a series of enzymes. The flavonoid biosynthetic pathway is regulated by many transcription factors, among which MYB transcription factors are thought to be key regulators. Hickory (*Carya cathayensis*) is an economic forest tree species belonging to the Juglandaceae family, and its fruit is rich in flavonoids. The transcriptome of exocarp and seed of hickory has previously been sequenced and analyzed by our team, revealing that CcMYB12 (*CCA0691S0036*) may be an important regulator of flavonoid synthesis. However, the specific regulatory role of CcMYB12 in hickory has not been clarified. Through a genome-wide analysis, a total of 153 *R2R3-MYB* genes were identified in hickory, classified into 23 subclasses, of which *CcMYB12* was located in Subclass 7. The *R2R3-MYB*s showed a differential expression with the development of hickory exocarp and seed, indicating that these genes may regulate fruit development and metabolite accumulation. The phylogenetic analysis showed that CcMYB12 is a flavonol regulator, and its expression trend is the same as or opposite to that of flavonol synthesis-related genes. Moreover, CcMYB12 was found to be localized in the nucleus and have self-activation ability. The dual-luciferase reporter assay demonstrated that CcMYB12 strongly bonded to and activated the promoters of *CcC4H*, *CcCHS*, *CcCHI*, and *CcF3H*, which are key genes of the flavonoid synthesis pathway. Overexpression of *CcMYB12* in *Arabidopsis thaliana* could increase the content of total flavonoids and the expression of related genes, including *PAL, C4H, CHS*, *F3H*, *F3’H*, *ANS*, and *DFR*, in the flavonoid synthesis pathway. These results reveal that CcMYB12 may directly regulate the expression of flavonoid-related genes and promote flavonoid synthesis in hickory fruit. Notably, the expression level of *CcMYB12* in hickory seedlings was significantly boosted under NaCl and PEG treatments, while it was significantly downregulated under acid stress, suggesting that CcMYB12 may participate in the response to abiotic stresses. The results could provide a basis for further elucidating the regulation network of flavonoid biosynthesis and lay a foundation for developing new varieties of hickory with high flavonoid content.

## 1. Introduction

A flavonoid is a compound with a C6–C3–C6 basic structure, consisting of two benzene rings and three carbon atoms. Flavonoid synthesis originates from p-coumaryl-coenzyme A, which is produced from phenylalanine ammonia-lyase (PAL), cinnamic acid 4-hydroxylase (C4H), and 4-coumaric acid-coenzyme A ligase (4CL) in the phenylpropane pathway. It is further converted into naringin under chalcone synthetase (CHS), chalcone isomerase (CHI), and other enzymes and serves as a major metabolite entering the synthetic pathways of other flavonoids via enzymes such as flavanone 3-hydroxylase (F3H), F3’H, flavone synthase (FNS), flavonol synthase (FLS), etc. [1,2]. In nature, there are many kinds of flavonoids, which can be divided into anthocyanins, flavanones, flavonoids, flavonols, flavanols, and so on based on their structural differences. Flavonoids exert an enormous function in plant growth and defense response [3]. Flavonoids also give plants a variety of colors, such as flavonols (white and pale yellow pigments) and anthocyanins (red, orange, blue, and purple pigments) [4]. As a kind of antioxidant, flavonoids have the ability to scavenge reactive oxygen species (ROS) and protect plants from various abiotic and abiotic stresses [5,6]. In addition to the benefits in plants, flavonoids have good health benefits in humans. It has been reported that the compound 3,7,3’,4’-tetrahydroxy flavone, a natural flavonoid commonly known as fisetin, can affect signaling pathways in different cancers, making it a potential drug for cancer prevention and treatment [7].

The expression of flavonoid biosynthesis-related genes is regulated by many transcription factors, consisting of basic helix loop helix (bHLH), WD40 repeat (WDR) factors, and MYB transcription factors [8,9,10]. Among them, MYB proteins have been shown to be key factors regulating flavonoid biosynthesis [11]. The MYB gene family is widespread in eukaryotes and has a highly conserved N-terminal DNA domain, which typically is composed of one to four incomplete amino acid sequence repeats (R). Each R sequence comprised approximately 52 amino acids(aa), producing three α helices [12]. According to the structural features of highly conserved domains, MYB transcription factors can be classified into four subclasses: 1R-MYB/MYB-related, R2R3-MYB, 3R-MYB, and 4RMYB (4 R1/R2 repeats) [13]. Among these, R2R3-MYB is the dominant subclass, which can regulate plant-specific processes, including development, secondary metabolism, stress response, and others [14].

The regulation of flavonoids by MYB transcription factors has been confirmed in various plants. However, different types of MYBs play different roles in regulating flavonoid synthesis. For example, PbMYB10B is an activator of anthocyanin and PA biosynthesis in the fruits of *Pyrus betulaefolia* [15]. CsMYB2 and CsMYB26 promote flavonoid biosynthesis in *Camellia sinensis* [16]. In *Gentiana scabra*, GtMYBP3 and GtMYBP4 enhance flavonoid biosynthesis in flowers at the early stage [17]. On the contrary, some MYB transcription factors have inhibitory effects on flavonoid synthesis. For instance, AtMYB4 inhibits flavonoid accumulation by inhibiting the expression of the AtADT6 gene in *Arabidopsis thaliana* [18]. In *Narcissus tazetta L*, NtMYB3 negatively regulates flavonoid biosynthesis by suppressing the biosynthesis of flavonols [19].

Hickory (*Carya cathayensis*), a woody oil nut tree in the Juglandaceae family, is widely cultivated in Zhejiang and Anhui Provinces, China [20]. Its nuts are rich in a variety of components, such as lipids, proteins, starches, sugars, vitamins, amino acids, mineral elements, small peptides, polyphenols, and flavonoids, with high nutrient and health values [21,22]. According to the *Compendium of Materia Medica* (a book on Chinese herbal medicine), hickory nuts can nourish the kidneys and lungs, nourish qi and blood, regulate dryness and heat, and relieve asthma [23]. Previous research found that flavonoids in the green exocarp and seeds of hickory have anticancer, antibacterial, and anti-aging properties [24]. Furthermore, the transcriptome of exocarp and seeds of *Carya cathayensis* were sequenced, and a co-expression network was constructed, showing that *CcMYB12* is co-expressed with some flavonoid-synthesis-related genes [25]. Previous studies have also shown that MYB12 is involved in the formation of flavonols in *Arabidopsis thaliana*, *Camellia sinensis*, *Lilium brownii*, and *Myrica rubra Sieb* [26,27,28,29]. However, how CcMYB12 regulates flavonoid biosynthesis in *Carya cathayensis* remains unknown.

In this study, we performed genome-wide analysis of all R2R3 family members of *Carya cathayensis* and assayed their expression profiles during exocarp and seed development. Meanwhile, we also identified and characterized *CcMYB12*. Using real-time quantitative PCR (RT-qPCR), the expression trends of *CcMYB12* and some genes related to flavonoid synthesis with the development of exocarp were analyzed. In addition, CcMYB12 was subjected to subcellular localization assay, transactivation assay, dual-luciferase reporter assay (DLRA), and functional verification via heterologous expression. Finally, the response of *CcMYB12* to abiotic stresses (salt, acid, and PEG treatment) was determined. Through these studies, we were able to clarify the specific regulatory function of CcMYB12 in flavonoid synthesis and provide theoretical guidance for future molecular breeding of *Carya cathayensis*.

## 2. Results

### 2.1. Identification, Phylogenetic Analysis, and Classification of R2R3-MYB Gene Family in Carya Cathayensis

Based on 126 R2R3-MYB proteins in *Arabidopsis thaliana*, 153 R2R3-MYB genes were identified in *Carya cathayensis* (Appendix A). A phylogenetic tree of these genes was constructed, showing that the R2R3-MYB genes of *Arabidopsis* and *Carya cathayensis* were divided into 23 subclasses, resembling previous studies (Figure 1a) [14]. From the phylogenetic tree, we observed that 115 *CcR2R3-MYB* belonged to S1 (5), S2 (4), S3 (1), S4 (7), S5 (1), S6 (2), S7 (8), S9 (7), S10 (4), S11 (5), S12 (4), S13 (3), S14 (14), S15 (4), S16 (1), S18 (6), S19 (1), S20 (7), S21 (7), S22 (6), S23 (2), S24 (4), and S25 (6) (Figure 1b). Of these, the subclass S14 had the most members, while S3, S5, S16, and S19 had the fewest. Notably, 38 *CcR2R3-MYB* genes were not classified.

### 2.2. Gene Structure and Protein Motif Assay of R2R3-MYB Family Members in Carya Cathayensis

To investigate the functional multiformity of CcR2R3 MYB proteins, conserved motifs were identified and analyzed by MEME (Figure 2a). A total of 10 conserved motifs (Motifs 1–10) were found, most of which were regularly distributed in the N-terminus, and a few irregularly distributed in the C-terminus. Almost all CcR2R3 MYBs contained more than two conserved motifs; most of them had one, three, four, or eight conserved motifs. It was also observed that CcR2R3 MYBs belonging to the same subfamily had similar conserved motifs, whereas there were different motifs between CcR2R3 MYBs from different subfamilies. For example, all CcR2R3 MYBs had one, two, three, four, six, or seven conserved motifs in S2. Compared with S4, S15 lacked conserved Motif 6. These results imply that these conserved motifs may be relevant to special effects of subgroups. To gain insight into gene function, regulation, and evolution, the exon–intron structure of 153 full-length CcR2R3-MYB genes was further analyzed (Figure 2b). Among these genes, the number of introns varied from zero to three, and more than half of them (110/153) had three exons. However, the gene structure showed only one or two exons in S22 and S23. These gene structure results are consistent with gene clustering by the phylogenetic tree, supporting the reliability of gene classification.

### 2.3. Expression Profiling of R2R3-MYB Family Members during Fruit Development in Carya Cathayensis

The developmental expression patterns of *153 CcR2R3-MYB* genes in different tissues (exocarp and seed) were analyzed based on previous transcriptome data measured by our team and are shown in Figure 3. In the exocarp, 51 of 153 *CcR2R3-MYB* genes had expression levels greater than 1 FPKM at the indicated developmental stages, among which 11 had the highest expression at the early stages, especially *CCA0598S0030*, with a value over 100, while 18 (mostly distributed in S20) appeared at the middle and late stages. During seed development, the expression level of 34 *CcR2R3-MYB* genes was greater than 1 FPKM at the sampling time points. Among them, the expression trend of 10 out 34 genes showed a gradual decrease, while 9 out of 34 showed a contrary trend. In all 23 subclasses, the expression levels of 6 genes in S22 were the highest and increased gradually with seed development. The expression of most genes in subclass S20 increased gradually during exocarp development and decreased gradually in developing seeds. These results suggest that CcR2R3-MYBs with high expression and fold changes during development might play an important role in fruit development, fruit ripening, metabolite accumulation (e.g., lipids, flavonoids, tannins, etc.), and environmental adaption.

### 2.4. Identification and Characterization of CcMYB12 and Its Potential Relationship with Flavonol Synthesis

Previous studies have shown that AtMYB12 is positively correlated with flavonol synthesis in Arabidopsis thaliana. Figure 1a shows that the CCA0691S0036 gene, in Subgroup 7 of the phylogenetic tree, had high homology with AtMYB12, so this gene was called CcMYB12. Furthermore, CcMYB12 CDS, which was cloned from the green exocarp of hickory fruits, encoded a protein of 459 amino acids with molecular weight of 50.86 kD and isoelectric point (PI) of 5.7. Phylogenetic analysis showed that CcMYB12 was evolutionarily related to flavonol regulators in Arabidopsis thaliana (AtMYB11, AtMYB12, and AtMYB111), Malus domestica (MdMYB22), Pyrus betulaefolia (PbMYB12b), and Vitis vinifera (VvMYBF1) (Figure 4a). Multiple sequence alignment indicated that transcription factors involved in flavonol biosynthesis from different species and CcMYB12 had conserved R2 and R3 domains, as well as SG7 motif, which is a flavonol-specific domain (Figure 4b). The N-terminus of these MYB transcription factors is highly conserved, but the C-terminus is diverse. 

Based on the KEGG database, flavonoid biosynthesis pathways in hickory were plotted (Figure 4c). Using RT-qPCR, the expression patterns of CcMYB12 and other genes, including CcPAL (CCA0633S0009), CcC4H (CCA0524S0150), CcCHS (CCA1379S0019), CcCHI (CCA1015S0108), CcF3H (CCA0705S0076), CcF3’H (CCA0507S0050), CcF3’5’H (CCA0795S0041), CcFLS (CCA0006S0007), and CcFNS (CCA0918S0074), involved in flavonol biosynthesis with exocarp development (S1, S2, S3, and S4) were further analyzed (Figure 4d,e). CcPAL showed a trend of first rising and then falling, and the maximum value appeared during Period S2. CcC4H and CcCHS had a similar expression from S1 to S4, showing a downward trend, with the maximum appearing during Period S1. CcF3H, CcF3’H, CcF3’5’H, and CcFLS showed similar expression patterns with exocarp development, displaying a trend of rising first and then falling, and the maximum value appeared in S3. The expression levels CcFNS did not change from S1 to S2, and then there was a sudden rise with about a six-fold change from S2 to S3, followed by a sharp decrease from S3 to S4. The expression trend of CcMYB12 was similar to that of CcC4H and CcCHS from S1 to S4 and contrary to that of CcF3H, CcF3’H, CcF3’5’H, and CcFLS from S1 to S3. These results suggest that CcMYB12 and some genes required for flavonol biosynthesis might have a co-expression relationship at some stages of fruit development.

### 2.5. CcMYB12 Is Localized in the Nucleus and Has Self-Activation Ability

After infiltrating constructs that produce 35S:CcMYB12-GFP fusion protein or 35S:GFP into tobacco leaves, the green fluorescence signal of 35S:CcMYB12-GFP was detected only in the nucleus, whereas 35S:GFP fluorescence alone was distributed throughout the cell (Figure 5a), suggesting that CcMYB12 is a nuclear localization protein. Using Y2H, the transactivation activity of CcMYB12 was also analyzed. The recombinant plasmid with CcMYB12 was successfully constructed, and then it and an empty vector were transferred into the yeast strain Y2H. Subsequently, we cultured these cells on SD/-Leu/-Trp medium and then screened them on tetradeficient medium (SD/-Leu/-Trp/-His/-Ade). It was observed that the yeast cells containing fusion or control plasmids grew on SD/-Leu/-Trp medium, but only the cells with fusion plasmid (CcMYB12-BD+AD) survived on SD/-Leu/-Trp/-His/-Ade medium (Figure 5b). These results suggest that CcMYB12 has self-activation ability.

### 2.6. CcMYB12 Activates the Promoter of Flavonoid Biosynthesis Genes

To further confirm the role of CcMYB12 in flavonoid biosynthesis, whether flavonoid-biosynthesis-related genes *CcC4H*, *CcCHI*, *CcF3H*, *CcF3’H*, *CcF3’5’H*, *CcANR*, *CcANS*, and *CcDFR* are targeted by CcMYB12 was investigated by dual-luciferase reporter assay (DLRA). To generate an effector construct, the CDS of CcMYB12 was cloned into pGreenII-62-SK (Figure 6a). Meanwhile, 1–2 kb promoter sequences upstream of the start code of *CcC4H*, *CcCHS*, *CcCHI*, *CcF3H*, *CcF3’H*, *CcF3’5’H*, *CcANR*, *CcANS*, and *CcDFR* were cloned into pGreenII0800-LUC, producing reporter constructs (Figure 6a). According to the known binding element of MYB12, the 2 kb promoter sequence upstream of the start code of these genes was selected to look for corresponding binding element. It was found that there were two, one, one, and one binding element in *CcCHS*, *CcF3H*, *CcF3’5’H*, and *CcDFR*, respectively, involved in flavonoid biosynthesis, while there were none in the other five genes, implying that CcMYB12 can bind to some genes (Figure 6b). DLRA showed that CcMYB12 strongly enhanced *CcF3H*, *CcC4H*, *CcCHI*, and *CcCHS* promoter activity compared with empty vector, while CcMYB12 did not directly activate *CcANS*, *CcDFR*, *CcANR*, and *CcF3’H* promoter (Figure 6c). Notably, *CcF3H* had the strongest activation, with a 34-fold change compared to control, while *CcF3’5’H* promoter was slightly activated and increased about 1.4-fold compared with control. These results suggest that CcMYB12 can directly activate some of the genes responsible for flavonoid biosynthesis, thus promoting the accumulation of flavonoids during hickory fruit development.

### 2.7. CcMYB12 Promoted the Expression of a Series of Flavonoid Biosynthesis Genes and Flavonoid Accumulation in Transgenic Arabidopsis Thaliana

To verify whether there were *CcMYB12* genes in the T3 of transgenic plants, three homozygous transgenic lines were identified. Figure 7a shows that *CcMYB12* genes were amplified in all three lines, except for wild-type *Arabidopsis* (Col-0). To select a suitable *Arabidopsis* overexpressing line, the expression level of *CcMYB12* in seeds of the three lines were analyzed. It was found that Line 2 had the highest expression level, which was 1.86-fold higher than Line 1 and 1.26-fold higher than Line 3 (Figure 7b). Therefore, Line 2 was used for further analysis. First, we measured the total flavonoids of seeds in transgenic Line 2 by UV spectrophotometer, which was about 1.7-fold higher than that of wild-type *Arabidopsis thaliana* (Figure 7c). By RT-qPCR analysis, it was found that the expression levels of seven genes related to flavonoid synthesis were increased, except for *At4CL*, in the seeds of transgenic *Arabidopsis*, and the expression level of *AtDFR* was the highest, with about a 14-fold increase (Figure 7d). These results suggest that CcMYB12 could promote flavonoid accumulation by directly regulating the expression of a series of flavonoid biosynthesis genes during fruit development in hickory.

### 2.8. CcMYB12 May Participate in Response to Abiotic Stress

The cis-regulatory elements were also predicted in the *CcMYB12* promoter (2 kb sequence upstream of ATG) and were mainly associated with light responsiveness (8), anaerobic induction (1), meristem expression (1), hormone response (3), and defense and stress response (2) (Figure 8a). To verify the role of *CcMYB12* in stress response, its expression level in hickory seedlings after treatment with salt stress, acid stress, and simulating drought stress for 6, 12, and 24 h was determined. Under salt stress, the expression level of *CcMYB12* increased gradually with time, showing 2.8-fold increments after 24 h (Figure 8b). Under acid stress, the expression of *CcMYB12* was significantly decreased at the indicated time points (Figure 8b). Under drought stress, the expression of CcMYB12 fluctuated, showing a tendency to first decrease, then increase, and then decrease (Figure 8b). These results reveal that CcMYB12 may be involved in the response to abiotic stresses and may play different roles in various stresses.

## 3. Discussion

*Carya cathayensis* is a woody oil nut tree in the Juglandaceae family [20]. Its exocarp and seeds contain rich flavonoids with anticancer, antibacterial, and anti-aging properties [24]. Based on previous transcriptome data of exocarp and seeds of *Carya cathayensis*, *CcMYB12* was shown to co-express with some flavonoid-synthesis-related genes [25]. Based on 123 *R2R3*-*MYB*s in *Arabidopsis thaliana*, 153 genes of the *Carya cathayensis* genome were searched, 115 of which were classified into 23 subgroups, and 38 were undivided according to the conservation of the N-terminal MYB domain and the C-terminal motif (Figure 1a), which agrees with the previous classification [14]. Members of the same subgroup usually share protein motifs and gene structures, as noted in Figure 2. 

Using previous RNA-seq data, it was found that the expression of most *CcR2R3*-*MYB*s changed with exocarp and seed development, accompanied by similar expression trends in the same subgroups (Figure 3). Previous studies also revealed that the members in the same subgroup share similar functions. For instance, AtMYB114, AtMYB113, AtMYB90, and AtMYB75 distributed in S6 are responsible for anthocyanin biosynthesis [30]. AtMYB52, AtMYB54, and AtMYB69 are distributed in S21, which can regulate cell-wall thickening in fibroblasts [31]. AtMYB11, AtMYB12, and AtMYB111 are found in S7, which are specifically involved in flavonol biosynthesis [32]. *CcA0691S0036* was identified in Subgroup 7 and is highly homologous to *AtMYB12*, and their protein sequence retains specific SG7 flavonol-specific motifs [33], similar to AtMYB11 and AtMYB12; thus, it is called *CcMYB12* (Figure 4a,b). In different plants, the function of the same MYB protein subgroup is widely conserved. It has been shown that MYB12 can regulate flavonol accumulation in *Arabidopsis thaliana*, *Camellia sinensis*, *Lilium brownii*, and *Myrica rubra Sieb* [26,27,28,29]. 

Using RT-qPCR, it was confirmed that the expression trend of *CcMYB12* was similar or opposite to that of flavonol-synthesis-related genes such as *CcCHS*, *CcCHI*, *CcF3H*, *CcF3’H*, *CcF3’5’H*, and *CcFLS* from S1 to S3 (Figure 4e), suggesting that *CcMYB12* may regulate flavonol accumulation by directly targeting these genes. The subcellular localization and self-activation analysis showed that CcMYB12 is a transcriptional activator (Figure 5). DLRA further displayed that CcMYB12 significantly activated flavonoid-synthesis-related genes, including *CcC4H*, *CcCHS*, *CcCHI*, and *CcCF3H*, while it had no effect on *CcF3’H*, *CcF3’5’H*, *CcANS*, *CcANR*, and *CcDFR* (Figure 6c), verifying the direct regulation of *CcMYB12* on some genes responsible for flavonol synthesis. Notably, the expression trends of *CcMYB12* and *CcCF3H* with exocarp development were the opposite from S1 to S3, and this was not consistent with the transcriptional activity assay. It was speculated that some novel regulators are involved in the direct regulation of *CcF3H* in hickory. Additionally, it was found that there were no binding elements for AtMYB12 in the promoter of *CcCHI*, which showed significant activation activity (Figure 6b), implying that there are some potential binding elements for CcMYB12. This phenomenon was the opposite for *CcDFR*, suggesting that CcMYB12 may only bind to the promoter or be completed by other transcriptional regulators. 

In order to study the regulatory effect of CcMYB12 on flavonoid biosynthesis, CcMYB12 was ectopically expressed in *Arabidopsis thaliana* based on the immature transgenic technology of *Carya cathayensis*. In overexpressing plants, the total flavonoid content was improved, and the transcription level of most flavonoid biosynthesis genes was upregulated, except for *At4CL* (Figure 7), thus further suggesting the direct regulatory role of CcMYB12. Interestingly, higher expression levels of *CcANS*, *CcANR*, and *CcDFR* in the CcMYB12 transgenic line were contrary to those of DLRA. It is likely that CcMYB12 regulates the expression of these genes through an indirect pathway and promotes their expression.

Salt stress, drought stress, and acid–base stress seriously affect the growth and development of *Carya cathayensis*. To adapt to harsh environments, plants have evolved coping mechanisms, such as secondary metabolite accumulation, including flavonoids and anthocyanins. Therefore, it is urgent to ameliorate the yield and quality of *Carya cathayensis* by modulating the genes connected with secondary metabolites. It has been reported that HPPBF-1, as an MYB transcription factor, binds to salt-tolerance-related proteins and can increase salt tolerance in plants [34]. The overexpression of *ThMYB13* gene in pitcher plants increases K+ uptake and decreases Na+ accumulation, thereby increasing salt tolerance [35]. The expression level of *CeqMYB4* in *Casuarina equisetifolia* was upregulated under salt treatment. As predicted, some stress-response elements were found in the promoter of *CcMYB12* (Figure 8a). After salt treatment for 6, 12, and 24 h, the expression of CcMYB12 in hickory seedlings was gradually increased, indicating that CcMYB12 might be involved in response to salt-stress treatment (Figure 8b). Under salt stress, it may improve flavonoid synthesis, helping plants to cope with environmental changes. CcMYB12 was significantly upregulated after 12 h of PEG treatment (Figure 8b), indicating that it may participate in drought response. It is worth noting that the expression of the *CcMYB12* gene was significantly downregulated under acid stress (Figure 8b), suggesting that CcMYB12 may negatively regulate acid stress in plants. Further work will be performed to reveal the molecular mechanism of CcMYB12 in the regulation of various abiotic stresses. 

Previous studies have revealed that DELLA protein, the key transcription factor in the gibberellin signaling pathway, is capable of combining with transcription factors MYB12/MYB111 in the flavonol synthesis pathway, promoting the binding of MYB12 to the promoter of downstream target genes in the flavonol synthesis pathway to stimulate their expression [36]. It has been reported that the expression of MYB11, MYB12, and MYB111, key transcription factors in flavonol biosynthesis, was inhibited by BES1 activated by brassinsteroid (BR), thus reducing flavonol accumulation [37]. In the promoter region of CcMYB12, some hormone response elements were observed (Figure 8a), suggesting that CcMYB12 may be involved in hormone response, an idea which needs to be verified in the future.

To sum up, a total of 153 genes were authenticated from the genome of *Carya cathayensis* based on *R2R3-MYB*s in *Arabidopsis thaliana*. These genes were divided into 23 subclasses, and the *CcMYB12* gene was distributed in subclass 7. These *R2R3*-*MYB*s showed different expression profiles in hickory exocarp and seed development. The cloned CcMYB12 is a protein composed of 459 amino acids with a size of 50.86 kD, and it has a systematic evolutionary relationship with flavonol regulators. In the process of hickory exocarp and seed development, the expression trends of CcMYB12 and some flavonoid synthesis genes were the same or opposite, further suggesting that there is a close relationship between CcMYB12 and flavonol-synthesis-related genes. CcMYB12 is localized in the nucleus and has self-activation ability. Transcriptional activation analysis showed that CcMYB12 could directly mediate the expression of some critical genes in the flavonoid biosynthesis pathway. The heterologous overexpression of *CcMYB12* in *Arabidopsis thaliana* could improve the content of total flavonoids and expression of flavonoid-synthesis-related genes. It was also noted that the 2 kb upstream promoter of *CcMYB12* from ATG contained some stress-related binding elements, and the expression level of *CcMYB12* was significantly increased in hickory seedlings after treatment with NaCl and PEG, while the trend was just the opposite under acid stress, implying a possible role for *CcMYB12* in abiotic stress response. These results show that *CcMYB12* positively regulates flavonoid accumulation during fruit development in *Carya cathayensis* and has potential effects under abiotic stress, offering a basis for molecular genetic breeding of *Carya cathayensis*.

## 4. Materials and Methods

### 4.1. Identification, Phylogenetic Analysis, and Sequence Alignment of R2R3-MYBs 

To identify the R2R3-MYB family genes in hickory, BLAST was used to seek sequences in the databases (PJU, www.juglandaceae.net, accessed on 15 September 2022) based on query sequences of 126 R2R3-MYB family proteins in *Arabidopsis* (TAIR, https://www.arabidopsis.org/, accessed on 15 September 2022) [38,39]. Phylogenetic analysis was performed by using MEGA 7.0 software, and 1000 repeated bootstrapping tests were performed to prove the statistical reliability of the phylogenetic tree. Multiple sequences were aligned by using DNAMAN 9.0 software.

### 4.2. Analysis of Gene Structures, Protein Motifs, Promoter Cis-Regulatory Elements, and Physiochemical Properties 

The Gene Structure Display Server 2.0 website [40] and the GFF3 file were used to complete genome structure visualization. From genome and annotation files, the amino acid sequences of CcR2R3-MYBs were extracted to predict the conserved motif by the online MEME Suite tool (MEME-Suite.org, accessed on 20 September 2022), which was visualized via TBTools. The PlantCARE website [41] was applied to predict cis-regulatory elements of a 2000 bp promoter sequence upstream of the *CcMYB12* gene. Physicochemical characterization of the CcMYB12 protein was performed by using ExPASy (https://www.ExPASy.org/, accessed on 20 September 2022).

### 4.3. Plasmid Construction

The vector was constructed by the homologous recombination method. The coding sequence (CDS) of CcMYB12 was amplified by using PCR and then cloned into targeted vectors pCAMBIA1300-GFP, pGreenII-62-SK, and pGBKT7-BD. To obtain a construct with promoter–luciferase reporter, promoters (~1–2 kbp) of some target genes from genomic DNA in hickory, such as CcF3H, CcF3’H, CcF3’5’H, CcANS, CcC4H, CcCHI, CcCHS, CcDFR, and CcANR, were subcloned into the pGreenII0800-LUC vector. Primer sequences for PCR are provided in Appendix A online.

### 4.4. Plant Material and Treatment 

According to the fruit development period, hickory fruits were separately sampled on 14 June, 22 June, 30 June, and 8 July 2022 from the orchard at Zhejiang Agricultural and Forestry University (119°43′42′′ E, 30°15′16′′ N, elevation: 38 m), separately labeled as S1, S2, S3, and S4. After that, the exocarp was immediately peeled off manually and then placed in liquid nitrogen or kept in a refrigerator at −80 °C.

Wild-type *Arabidopsis thaliana* (Col-0) was grown under controlled environmental conditions (23 °C, 16 h light/8 h darkness, and 70% relative humidity). After flowering, the pCAMBIA1300-*CcMYB12*-GFP construct was impregnated by the flower soaking method [42], and the transformation process was repeated after 1 week. The seeds were collected and screened on hygromycin-supplemented MS medium until a homozygous T3 generation was obtained. The overexpressing *CcMYB12* lines were selected for further analysis according to RT-qPCR data.

Then 3-month-old well-growing seedlings of *Carya cathayensis* were treated with abiotic stress, including NaCl (150 mM), acid (pH = 3.5), and PEG (5%) treatment. The third mature leaves from the top down were collected at indicated time points (0, 6, 12, and 24 h). The samples were immediately put into liquid nitrogen, then milled, and RNA was extracted. Three biological replicates were prepared for each treatment. 

### 4.5. RNA Extraction and Real-Time Quantitative PCR (RT-qPCR) Analysis 

A rapid RNA isolation kit (Yueyang, China) was used to extract total RNA of samples. Based on the manufacturer’s instructions, cDNA was obtained with a PrimeScript^TM^ 1st strand cDNA Synthesis Kit (Takara, Gunma, Japan). Primers for gene expression assay were designed by Primer 5 software online, as shown in Appendix A. RT-qPCR was performed by using SYBR Green Premix reagent (Applied Biosystems, Waltham, MA, USA) and a CFX 96 real-time system (Bio-Rad, Hercules, CA, USA) on the basis of the manufacturer’s instructions. *AtACTIN*-*2* and *CcACTN* were used as internal criteria for normalization. The reaction process was 40 cycles at 95 °C for 10 s and 55 °C for 30 s, and 2^−ΔΔCT^ was used to reckon the relative expression levels of genes. The experiments were repeated three times.

### 4.6. Determination of Total Flavonoid Content

The total flavonoid content was determined based on the way described by Rosa et al. [43], with slight embellishments. First, 100 mL of sample extract was put into a 10 mL centrifuge tube, and then 1 mL of sodium nitrite (5%, NaNO_2_, *w/v*) was added. After mixing, it was reacted in the dark for 5 min. Then 1 mL of 10% aluminum chloride (AlCl_3_, *w/v*) was added for a further 3 min reaction. After adding 5 mL of 4% sodium hydroxide (NaOH, *w/v*) and reacting for half an hour under a dark environment, the absorption value at 510 nm was detected by using an ultraviolet–visible spectrophotometer (UV-2600, Shimadzu, Tokyo, Japan). The standard curve was plotted on the basis of (+)-catechin standard, and then total flavonoid levels were calculated.

### 4.7. Subcellular Localization Analysis of CcMYB12

*Agrobacterium tumefaciens* mediated transient expression in Nicotiana benthamiana leaves was used to detect the subcellular localization of proteins [44]. Once vectors were successfully constructed, they were transformed into A. tumefaciens strain GV3101-competent cells. These cells were screened on LB solid medium containing 50 μg mL^−1^ rifampicin (Rif), 50 μg mL^−1^ kanamycin (Kana), and 50 μg mL^−1^ gentamicin (Geta) in an incubator with a constant temperature of 28 °C. After 48 h, a positive colony was picked and put into liquid LB medium and cultured in a shaking incubator for 48 h, under a constant temperature of 28 °C. To obtain resuspension cells with an OD600 value of 1.0, the cell cultures (OD600 = 0.5–0.6) were centrifuged at 5000 rpm at room temperature for 10 min and then resuspended in MMA buffer (10 mM MES, 10 mM MgCl_2_, and 150 μM acetosyringone, pH 5.6). Then the cell cultures were incubated at room temperature in the dark for 2–3 h. Using a 1 mL syringe, the cultures were injected into health-growing N. benthamiana leaves. After 48 h, GFP fluorescence was determined under a laser confocal fluorescence microscope (excitation, 488 nm; emission, 495–515 nm) (LSM 800, Zeiss, Oberkochen, Germany). The three repeats were conducted.

### 4.8. Transactivation Activity of CcMYB12 

Yeast two hybrid (Y2H) was used to detect the self-activation of CcMYB12 in yeast cells. Recombinant plasmid *pGBKT7*-*CcMYB12* and empty vector *pGADT7* were transferred into Y2H competent cells. The cells were first grown on SD/-Leu/-Trp medium for 2–3 days and then screened on SD/-Leu/-Trp/-His/-Ade tetradeficient medium.

### 4.9. Dual-Luciferase Reporter Assay 

Based on the protocol of the dual-luciferase reporter assay (DLRA) system, transcriptional activation analysis of CcMYB12 was completed [38]. pGreenII-62-SK-CcMYB12 and pGreenII0800-LUC-*CcF3H*, *CcF3’H*, *CcF3’5’H*, *CcANS*, *CcC4H*, *CcCHI*, *CcCHS*, *CcDFR*, and *CcANR* promoter constructs were transformed to *Agrobacterium GV3101/Psoup* strains, which were cultured in the same means for analysis of subcellular localization. Cells with OD600 = 0.6–0.8 were sampled for centrifugation and resuspended in MMA buffer to obtain suspensions with OD600 = 1.0. After that, these cells were blended according to the ratio of effector and reporter constructs (9:1). After 2–3 h of rest at room temperature in the dark, the mixed cells were injected into *N. benthamiana* leaves with a 1 mL syringe. After infiltrating for 48 h, the leaves were collected and ruptured with lysis buffer. Using a GLO-MAX 20/20 luminometer (Promega, Madison, WI, USA), the firefly luciferase activity (LUC) and Renilla luciferase activity (REN) of samples were determined in turn. Final transcriptional activity was calculated based on the ratio of LUC to REN. The experiments were repeated three times.

### 4.10. Statistical Analysis 

A statistical assay was conducted by one-way analysis of variance (ANOVA), using SPSS 21 (SPSS Inc., Chicago, IL, USA). Significances among treatments were determined by *p*-values (Duncan test or t-test, *p* < 0.05).

## Figures and Tables

**Figure 1 ijms-23-15618-f001:**
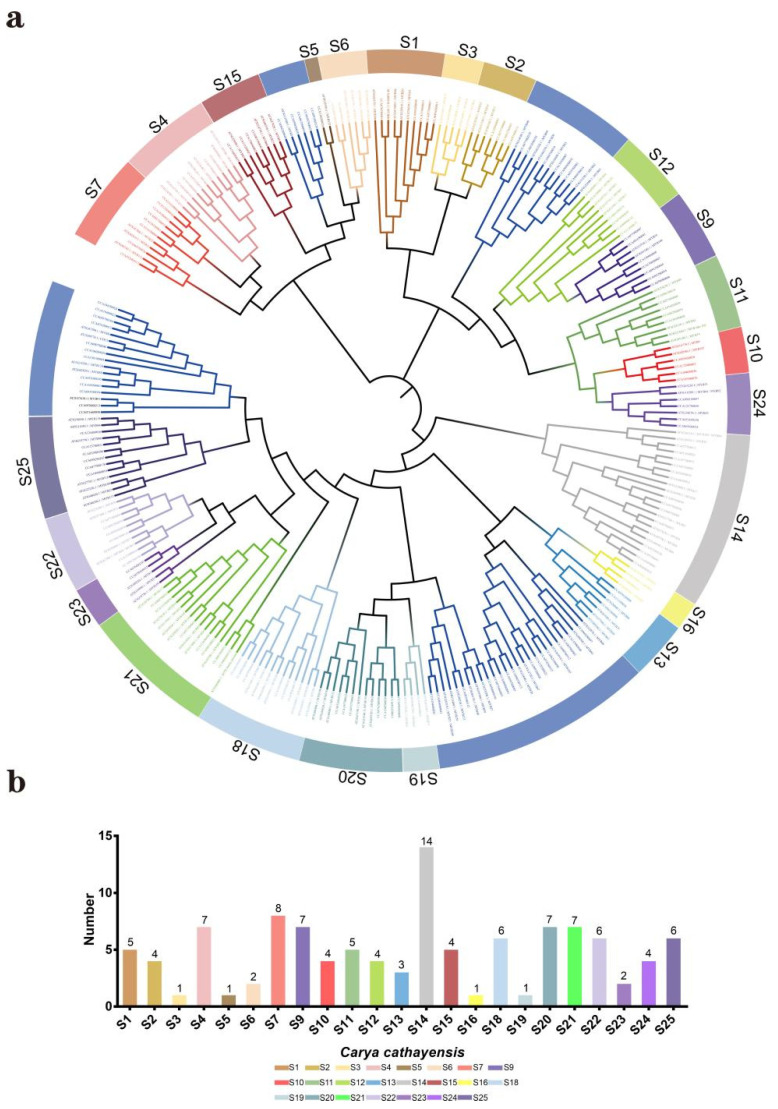
Genome-wide identification and classification of R2R3-MYB family genes in *Carya cathayensis*. (**a**) Phylogenetic tree of hickory and *Arabidopsis R2R3-MYB* family genes generated by neighbor joining (NJ) method in MEGA 7.0. (**b**) Quantitative statistics of *CcR2R3-MYB* family genes in different subgroups. The number above each column represents the number of genes in each subclass.

**Figure 2 ijms-23-15618-f002:**
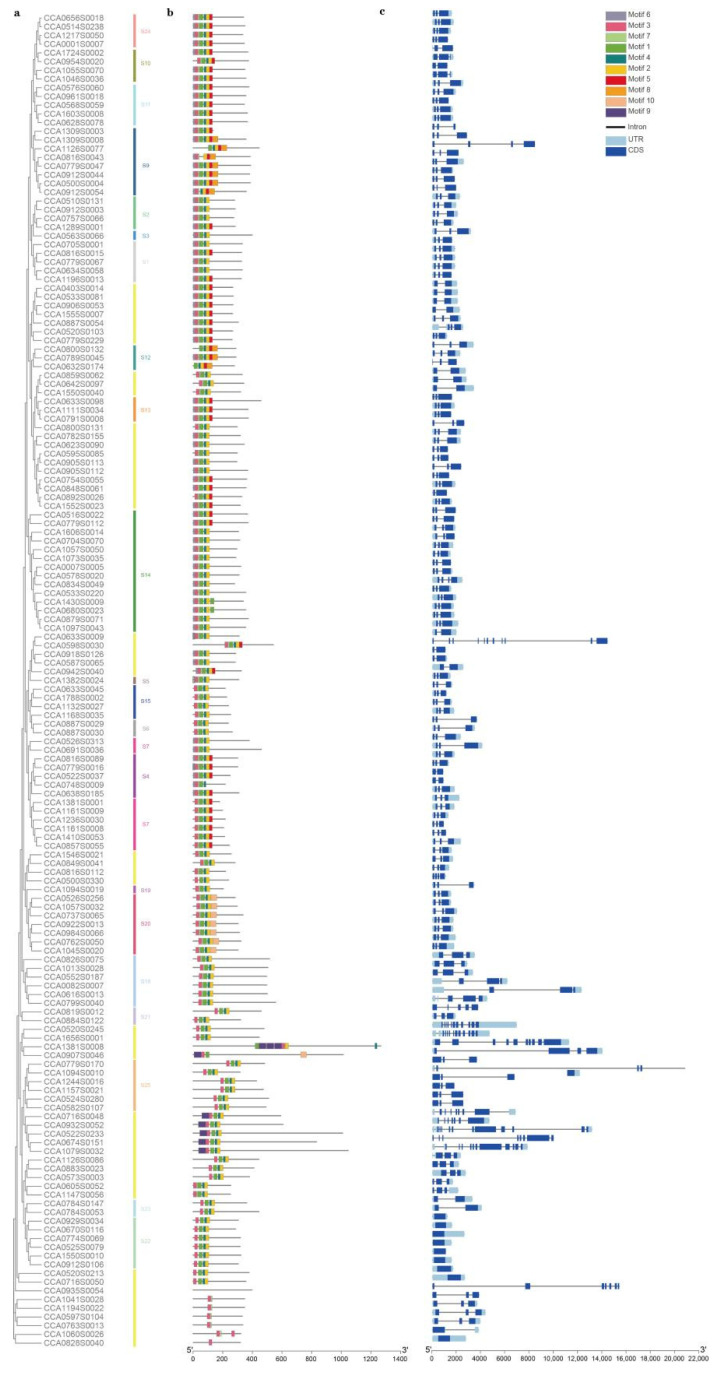
Phylogenetic analysis, protein conservative motifs, and gene structure of R2R3-MYB family genes in *Carya cathayensis*. (**a**) Phylogenetic trees were established according to full-length aa sequence alignment of 153 R2R3-MYBs in *Carya cathayensis*. Solid-colored vertical lines indicate 23 subgroups. (**b**) Schematic of protein conservative motifs. Solid colored boxes represent protein motifs. (**c**) Gene structure display of *R2R3-MYB* family genes in *Carya cathayensis*. Solid black line and blue box indicate introns and exons, respectively.

**Figure 3 ijms-23-15618-f003:**
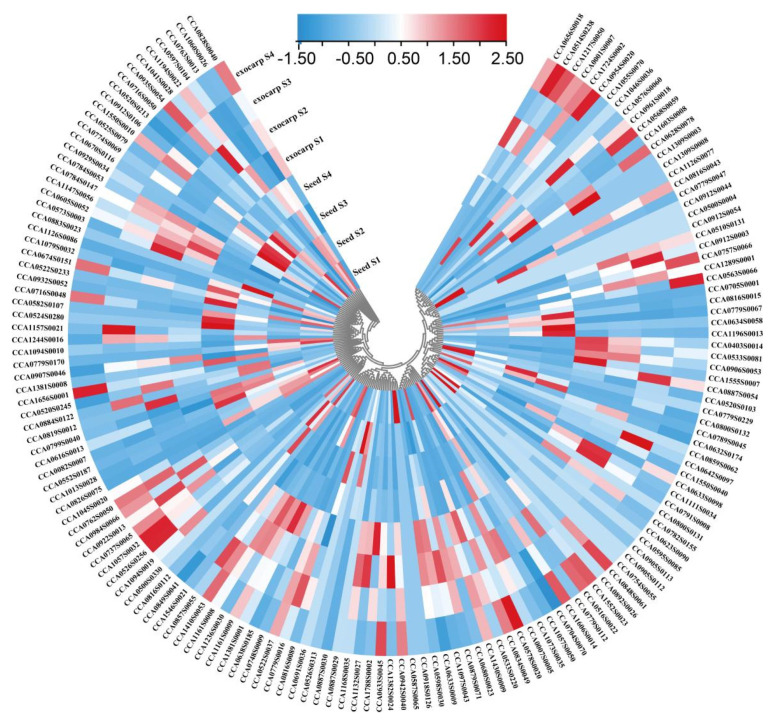
Expression pattern analysis of all *R2R3*−*MYB* family genes during hickory exocarp and seed development. A hierarchical clustering heatmap was plotted based on FPKM via log2 transformation. Color bars indicate range of log2 (FPKM).

**Figure 4 ijms-23-15618-f004:**
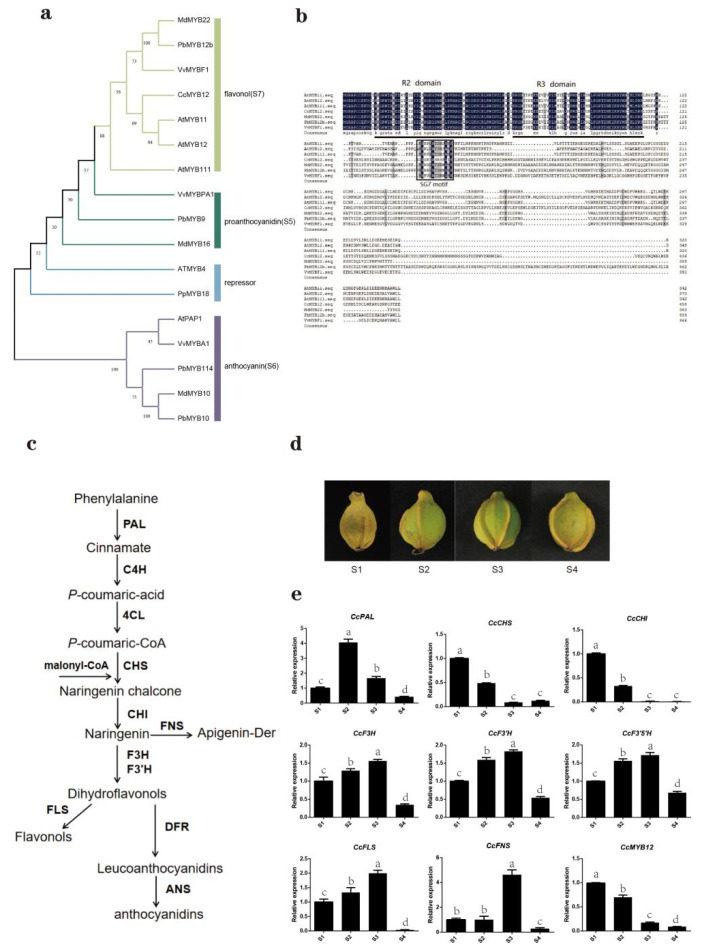
Relationship between CcMYB12 and flavonol accumulation in *Carya cathayensis*. (**a**) Phylogenetic analysis of CcMYB12 and R2R3 transcription factors regulating flavonoids, generated by NJ method in MEGA 7.0. (**b**) Protein sequence alignment of CcMYB12 and R2R3-MYB transcription factors from Subgroup 7. Positions of R2 and R3 MYB domains are indicated by black bar below alignment. SG7 domain is highlighted by a black box. (**c**) Plotted diagram of flavonoid synthesis pathway based on KEGG data. (**d**) Developing fruits of *Carya cathayensis*. Sampling time points noted as S1, S2, S3, and S4. (**e**) Relative expression levels of flavonoid-synthesis-related genes and *CcMYB12* in hickory exocarp at indicated time points. Different letters in the error bar indicate obvious differences among samples, as determined by ANOVA (Duncan test, *p* < 0.05).

**Figure 5 ijms-23-15618-f005:**
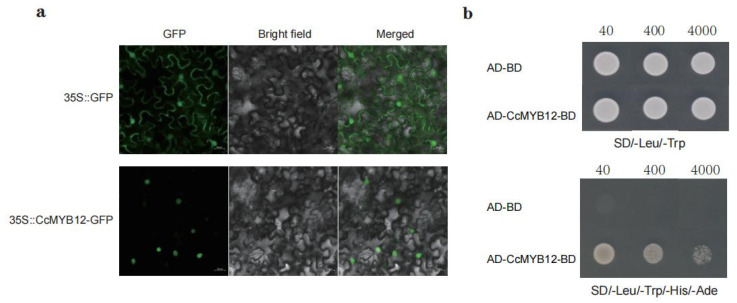
CcMYB12, located in nucleus, has self-activation ability. (**a**) Subcellular localization observation of CcMYB12 in tobacco system with empty GFP vector as control. (**b**) Verification of CcMYB12 self-activation ability, as tested by Y2H. Transformed yeast cells were cultured on SD/-Leu/-Trp-deficient medium and SD/-Leu/-Trp/-His/-Ade-deficient medium to test their activity.

**Figure 6 ijms-23-15618-f006:**
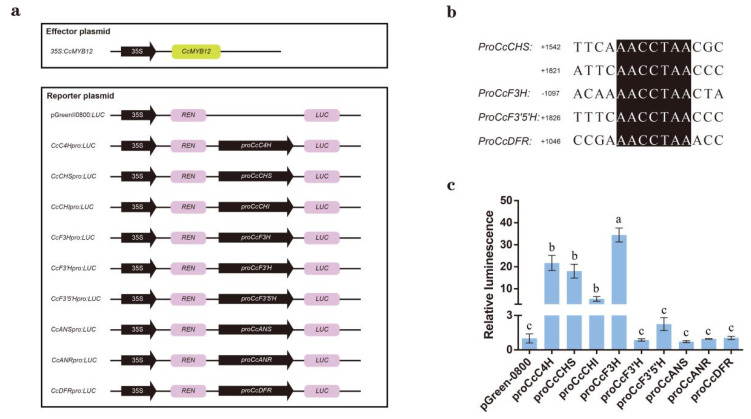
CcMYB12 can directly activate some genes involved in flavonoids synthesis. (**a**) Schematic diagram of construction of effector and promoter for DLRA. (**b**) Binding element analysis of promoters of *CcC4H*, *CcCHI*, *CcF3H*, *CcF3’H*, *CcF3’5’H*, *CcANR*, *CcANS*, and *CcDFR* based on AtMYB12. (**c**) Transcriptional activation analysis of CcMYB12 on *CcC4H*, *CcCHI*, *CcF3H*, *CcF3’H*, *CcF3’5’H*, *CcANR*, *CcANS*, and *CcDFR*. LUC and REN of samples were determined sequentially with a GLO−MAX 20/20 photometer (Promega, USA), and the ratio of LUC and REN was taken as transcriptional activity. The data from 3 biological replicates are presented as mean ± SE. Various letters in error bar indicate obvious differences among distinct groups (Duncan test, *p* < 0.05).

**Figure 7 ijms-23-15618-f007:**
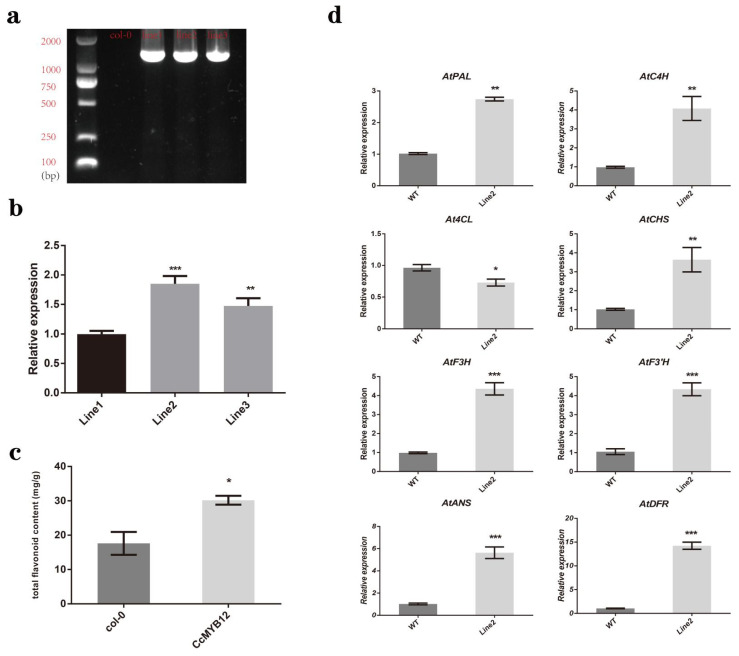
CcMYB12 can increase total flavonoid content and regulate some genes required for flavonoid synthesis. (**a**) Homozygous T3 *Arabidopsis* seeds contained *CcMYB12* gene. (**b**) *CcMYB12* genes showed different expression levels in three transgenic lines. (**c**) *CcMYB12* overexpressing line had higher total flavonoid accumulation than wild type (Col-0). (**d**) Expression levels of most flavonoid-synthesis genes, including *CcPAL*, *CcC4H*, *CcF3H*, *CcF3’H*, *CcANR*, *CcANS*, and *CcDFR*, were increased in the transgenic line compared to the wild type. Asterisks in error bars indicate significant differences among different groups (t-test, *p* < 0.05): *** extremely significant (*p* < 0.01); ** very significant (0.001 < *p* < 0.01); * significant (0.01 < *p* < 0.05); and ns, not significant (*p* > 0.05).

**Figure 8 ijms-23-15618-f008:**
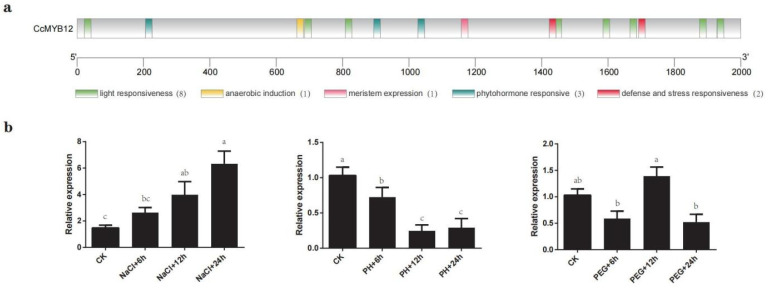
CcMYB12 may be involved in salt, acid, and PEG response. (**a**) Analysis of cis-acting elements in the promoter region of *CcMYB12*. Solid-colored boxes represent types of cis-acting elements. (**b**) Expression levels of *CcMYB12* under different abiotic treatments. Gene-expression level was calculated based on the 2^−ΔΔCT^ method, and *CcACTIN* was used as an internal reference for normalization. Different letters in the error bar denote significant differences among different groups (Duncan test, *p* < 0.05).

## Data Availability

Not applicable.

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
