# Peer review of "CcMYB12 Positively Regulates Flavonoid Accumulation during Fruit Development in Carya cathayensis and Has a Role in Abiotic Stress Responses"

_ijms, 2022, doi:10.3390/ijms232415618_

Round 1
Reviewer 1 Report
The manuscript entitled „CcMYB12 positively regulated flavonoid accumulation during fruit development in Carya cathayensis as well as their roles in abiotic stress responses” is a well-written text presenting well-designed research. The topic of research, which is the expression of genes and their regulation, is still timely due to the complexity of the mechanisms and the need for an in-depth study of the processes controlling it on various examples. The case when gene expression is associated with the production of substances important also from the point of view of human use, adds additional value to the presented results.
The theoretical background and the object of research was characterized comprehensively, introducing a reader to the subject. The purpose of the research was clearly formulated. The methodology has been correctly selected and comprehensively described. The results are clearly presented, graphics adequate to the text, well illustrate the content. Discussion of the results is comprehensive and refers to up-to date literature.
I recommend accepting the manuscript for publication in IJSM.
There are only minor cosmetic corrections which could help to improve the manuscript:
1. Please, check the grammar in the sentence starting in line 384 “Further work…”
2. The manuscript should be checked for double spaces, which are present for example in lines: 207, 223, 259, 324, 375, 504. And lack of space – e.g. 167, 275, 370.
3. Please, check and unify the temperature records (fonts and spaces, E.E LINE 445) and Greek symbols (line 69).
4. Check for double full stops.
5. I would also advise to use a bigger size of font in Figure 4a, if it is possible.
Reviewer 2 Report
The article "CcMYB12 positively regulated flavonoid accumulation during fruit development in Carya cathayensis as well as their roles in abiotic stress responses" is devoted to a comprehensive analysis of the novel transcription factor CcMYB12 from hickory (Carya cathayensis) and its role in flavonoid metabolism. The authors of the study rely on previous work and proven verified approaches. By analyzing 153 hickory transcription factor genes, the research team discovered and characterized a transcription factor belonging to the MYB family, CcMYB12. This TF is a homologue of the Arabidopsis transcription factors AtMYB12. A wide range of bioinformatics tools were used for the search. However, the authors did not limit themselves to this and analyzed open TF using a two-hybrid system, DLRA, and a number of other biochemical and molecular biological (qPCR) approaches to fully and comprehensively determine the role of this transcription factor, abioitic stress response involvement and the metabolic pathways in which it is included.
Among the special advantages it is worth noting:
- beautiful illustrations
- the article is very well structured
Along with this, the work is not without some minor shortcomings:
- Figure 4 has insufficient resolution, which makes small details difficult to distinguish (perhaps this problem is specific only to the manuscript, but not to the final publication).
- the sentence "According to the record in ... and relieve asthma" (page 3, lines 89-91) would be better rewritten relying on evidence-based medicine or more general way.
However, these remarks in no way spoil the impression of this study and it can be published after minor revisions.
